# Dense-Frequency Signal-Detection Based on the Primal–Dual Splitting Method

**DOI:** 10.3390/e24070991

**Published:** 2022-07-18

**Authors:** Jiaoyu Zheng, Zheng Liao, Xiaoyang Ma, Yanlin Jin, Huangqi Ma

**Affiliations:** The College of Electrical Engineering, Sichuan University, Chengdu 610065, China; zhengjiaoyu@stu.scu.edu.cn (J.Z.); liaozheng@stu.scu.edu.cn (Z.L.); jinyanlin@stu.scu.edu.cn (Y.J.); mahuangqi@stu.scu.edu.cn (H.M.)

**Keywords:** harmonic, interharmonic, dense-frequency signal, entropy, phase analysis, primal–dual splitting method

## Abstract

Aiming to solve the problem of dense-frequency signals in the power system caused by the growing proportion of new energy, this paper proposes a dense-frequency signal-detection method based on the primal–dual splitting method. After establishing the Taylor–Fourier model of the signal, the proposed method uses the sparse property of the coefficient matrix to obtain the convex optimization form of the model. Then, the optimal solution of the estimated phasor is obtained by iterating over the fixed-point equation, finally acquiring the optimal estimation result for the dense signal. When representing the Taylor–Fourier model as a convex optimization form, the introduction of measuring-error entropy makes the solution of the model more rigorous. It can be further verified through simulation experiments that the estimation accuracy of the primal–dual splitting method proposed in this paper for dense signals can meet the M-class PMU accuracy requirements.

## 1. Introduction

In recent years, with the promotion of renewable-energy-generation technology, an increasing number of distributed energy resources has been employed, leading to the generation of numerous harmonic and interharmonic signals [1]. When these harmonic and interharmonic signals are of similar frequencies, spectrum aliasing often occurs, damaging the power system. Therefore, it is necessary to quickly and accurately detect the harmonic and interharmonic signals in the power system and manage them.

Currently, harmonic and interharmonic signal-detection methods can be divided into three categories. Methods of the first category are based on the discrete Fourier transform. The discrete Fourier transform method is simple to implement and has low computational complexity [2]. It is a mature theory, and it is suitable for static signals. However, there are often deviations in the harmonic parameters using DFT, which are mainly due to the synchronization deviation of signal sampling, the fence effect, and spectral leakage. The Taylor–Fourier Transform algorithm (TFT) assumes a fluctuating, compound envelope line at each harmonic, and the algorithm can estimate the time-varying harmonics within the observation window, suffering less from distortion and allowing less harmonic interference [3]. The Taylor weighted least squares method (TWLS) can reduce harmonic penetration and noise interference, but it is strongly disturbed by the second harmonic [4]. To overcome the shortages of the DFT method, various improved methods have been proposed. Common techniques used in these methods include tracking the fundamental frequency and adding window interpolation or spectral correction. The synchronization offset can be reduced by adjusting the sampling strategy. For example, the zero-crossing detection algorithm, the phased-locked loop algorithm, and the time-domain quasi-synchronous sampling algorithm based on Newton’s interpolation (TQSA) obtain the truncated full-cycle signal whenever possible to maintain synchronous sampling [5,6,7]. However, these methods are only applicable to certain conditions and are not capable of conducting accurate synchronous sampling for dense and complex signals. Ref. [8] proposed the DFSD method to process dense signals. Besides that, spectral leakage can be solved by introducing a window function, and the fence effect can be solved by interpolating the signal. Therefore, various methods of window interpolation have been proposed, such as the rectangular window, Hanning window, Nuttall window, and Triangular self-convolution window [9,10,11,12]. However, these window functions increase the side-flap attenuation at the expense of the main-flap width, limiting the range of frequency measurements. To make the mainlobe width as small as possible, reduce the sidelobe attenuation rate quickly, and further suppress the spectrum leakage, scholars have combined the quasi-synchronous sampling method with window interpolation. The main methods include Newton’s interpolation, interpolated FFT (IpFT), and chirp Z-transformation [13,14]. However, when one of the main spectral lines of the signal is dominated by the main spectral lines of adjacent frequencies, the above-mentioned synchronous sampling, windowing interpolation, and spectral correction methods become rather ineffective or even fail.

The second class of methods are the modern spectral estimation methods. Full-period sampling is not a requirement for this type of method, which fundamentally avoids the inherent flaws of DFT, and which has a high level of detection accuracy. The MUSIC algorithm, including its improved algorithms, such as iMUSIC and real-valued MUSIC, as well as the Prony algorithm, including Subsampling Prony, are representatives of modern spectral estimation algorithms [15,16,17,18,19]. These methods have high accuracy and resolution of signal estimation and good stability of spectral estimation. However, these methods require a large number of calculations and are susceptible to interference from noises, and the uncertainty of the detection results will increase when the number of harmonics detected is too large. The rotational invariance estimation signal parameter algorithm (ESPRIT) has no limit for sampling quantities, but it requires high accuracy in modeling and has a high computational complexity [20]. The matrix pencil algorithm and Instantaneous Spectral Analysis are less affected by noises and have better statistical properties for estimating frequency components without any limitations on the number of frequency components to be detected [21,22]. Still, these methods demand a lot of calculations. The chirp Z-transform algorithm provides excellent frequency resolution by computing the transformation of a finite-duration signal, along with having a general helical profile in the z-plane [23]. The decentralized energy distribution (DED) algorithm realizes the estimation of separate interharmonic components by calculating frequency deviations from the energy distribution in the spectrum [24]. MSSM samples the signal asynchronously and treats all of its components as interharmonics; the method identifies the peaks in the spectra obtained by DFT, thus separating the spectral content of each peak and estimating the parameters of each component [25]. Nonetheless, when measuring wideband, multi-frequency signals, the targeted spectrum may be merged into another spectrum, limiting the application of MSSM. The variational mode decomposition method (VMD) divides the input signal into a discrete number of sub-signals with sparse characteristics by assuming that each sub-signal is compacted around the center frequency [26]. The main advantages of this method are its ability to detect interharmonics and estimate time-dependent spectral components. However, parameters such as bandwidth and the number of frequency components to be extracted from the signal need to be defined. In summary, the estimation order of these methods will limit the number of estimated frequency components, and these methods cannot be specified to detect or calculate a certain frequency component. They can only be used in the full-frequency domain under the limitation of the order, which is less efficient and more computationally intensive. Therefore, the application of this type of method is very limited.

The methods in the third category combine traditional phase-estimation methods with machine learning algorithms, and this type of method can effectively reduce computational complexity. Typical algorithms include the genetic algorithm, which neither relies on information about the components of the objective function nor requires a priori conditions [27]. However, if the parameters of the objective function have a high correlation, the efficiency and reliability of the algorithm will be greatly confined. The neural network method has a low sensitivity to data, good real-time performance, and strong anti-interference ability [28]. Typical improved neural network methods are summarized below. The backward propagation network algorithm (BPN) enables error backpropagation training [29]. However, due to its multilayer structure and the greedy nature of the backpropagation algorithm, the algorithm has the problem of local minima in the error plane and slow convergence. The radial basis function neural network (RFNN) is simple in structure, but the algorithm does not have a standardized neural network construction method, and its accuracy is highly dependent on the samples [30]. The Kalman filter decomposition method and the extended Kalman filter decomposition method can effectively avoid spectrum leakage, but the state matrix of the Kalman filter is difficult to determine, and the methods have poor convergence speed and stability, limiting their application [31]. In reference [32], the minimum error entropy algorithm with sparse penalty constraint is proposed. By using the minimum error entropy criterion (MEE), the performance of the algorithm under non-Gaussian noise interference is improved, and the estimation model is as close as possible to the theoretical model. The particle swarm algorithm (PSOPC) mixes passive aggregated particle swarm with least squares, which has been used to effectively investigate the nonlinear phase and linear amplitude of harmonics [33]. Harmonic estimation based on ESPRIT-assisted adaptive wavelet neural network has fast convergence, fewer calculations, and higher detection accuracy [34]. The bacterial foraging optimization algorithm (BFO) can be used to solve non-gradient optimization problems [35]. The LS-based hybrid Firefly algorithm (FA-LS) can fully estimate the harmonic content in the tilted power signal, but a priori conditions are needed to update the data [36].

This paper uses the primal–dual splitting method to detect dense signals. The method first establishes a TFM model of the dense signal and performs a spectral analysis of the acquired signal. Considering the weak ability of the TFM model to identify interharmonics, the CSTFM model is introduced. The optimal estimate ξ of the CSTFM model is defined and so is the threshold value η of ξ. The solution of the CSTFM model is transformed into a convex optimization problem, and the accuracy of the phase estimation can be improved by iteration using the primal–dual splitting algorithm. When representing the model as a convex optimization form, the measurement error entropy is introduced to indicate the effect of various uncertainties in the measurement process. Simulation results have shown that the algorithm has relatively low computational complexity, high accuracy, and strong anti-interference capability for the estimation of dense signals, demonstrating its superiority compared with the other algorithms tested in this paper.

## 2. Estimation of Dense-Signal Harmonic Phase

Dense-frequency signals refer to harmonics or interharmonics that are very close together in the frequency domain. Supposing the signal has a total of *k* spectral lines after the fast Fourier transform, the spectral peaks for the $k_{1},k_{2},\ldots$ spectral line are observed, respectively, and the spacing between adjacent spectral peaks is $d=k_{n+1}-k_{n}$. Usually when $\left|d\right|\leq 2$, it can be considered that the adjacent harmonics have mainlobe interference, and the signal can be considered as a dense signal.

The static signal model x(t) in a power system can be expressed as:(1)$$x(t)=A_{h}\cos(2\pi f_{h}t+\varphi_{h})-\frac{T}{2}\leq t\leq \frac{T}{2}$$
where $A_{h}$ denotes the amplitude of the signal, $f_{h}$ denotes the frequency of the signal, and $\varphi_{h}$ denotes the phase of the signal.

The signal can be expressed in phase form as:(2)$$x_{h}(t)=A_{h}e^{j(2\pi f_{h}t+\varphi_{h})}$$

Sampling the signal at interval Δt, Equation (2) can be expressed as:(3)$$X_{h}(n\Delta t)=A_{h}e^{j(2\pi f_{h}n\Delta t+\varphi_{h})}$$
where $\Delta t=1/f_{s}$, $f_{s}$ is the sampling frequency, and n=0,1,2,…,N−1, N is the number of sampling points. Performing the discrete Fourier transform on the signal, Equation (3) can be transformed into (4):(4)$$X_{h}(k\Delta f)=\frac{1}{N}\sum_{n=0}^{N-1}A_{h}e^{j(2\pi f_{h}n\Delta t+\varphi_{h})}e^{-j(\frac{2\pi kn}{N}})$$
where Δf=1/(NΔt) is the frequency resolution.

When $f_{s}\gg f_{h}$, from the definition of the Riemann integral, Formula (4) can be expressed in the following form:(5)$$X_{h}(k\Delta f)=\frac{A_{h}e^{j\varphi_{h}}}{2\pi j}\frac{e^{j2\pi (f_{h}/\Delta f)}-1}{\left(f_{h}/\Delta f-k\right)}$$

The equation can be simplified as:(6)$$X_{h}(k)=\frac{\alpha_{h}}{\beta_{h}-k}$$
where:(7)$$\alpha_{h}=\frac{A_{h}e^{j\varphi h}}{2\pi j}(e^{j2\pi (fh/\Delta f)}-1)$$
(8)$$\beta_{h}=f_{h}/\Delta f$$

Assuming that there are m frequency components at position k of the signal spectrum, the dense-frequency model can be expressed as:(9)$$\begin{array}{ll} X(k) & =X_{1}(k)+X_{2}(k)+\ldots +X_{m}(k)\\ & =\frac{\alpha_{1}}{\beta_{1}-k}+\frac{\alpha_{2}}{\beta_{2}-k}+\ldots +\frac{\alpha_{m}}{\beta_{m}-k} \end{array}$$

According to Formula (9), Equation (10) can be obtained by the spectrum analysis of the sampled signal:(10)$$\left[\begin{array}{c} X(k_{1})\\ X(k_{2})\\ \vdots \\ X(k_{m}) \end{array}\right]=\left[\begin{array}{cccc} \frac{1}{\beta_{1}-k_{1}} & \frac{1}{\beta_{2}-k_{1}} & \ldots & \frac{1}{\beta_{m}-k_{1}}\\ \frac{1}{\beta_{1}-k_{2}} & \frac{1}{\beta_{2}-k_{2}} & \ldots & \frac{1}{\beta_{m}-k_{2}}\\ \vdots & \vdots & \vdots & \vdots \\ \frac{1}{\beta_{1}-k_{m}} & \frac{1}{\beta_{2}-k_{m}} & \ldots & \frac{1}{\beta_{m}-k_{m}} \end{array}\right]\left[\begin{array}{c} \alpha_{1}\\ \alpha_{2}\\ \vdots \\ \alpha_{m} \end{array}\right]$$

The above formula can be written as:(11)X=Bθ
where X is the sampled signal, $X={\left[X(k_{1}),X(k_{2}),\ldots ,X(k_{m})\right]}^{T}$, θ is a column vector containing Taylor–Fourier coefficients, $\theta ={\left[\alpha_{1},\alpha_{2},\ldots ,\alpha_{m}\right]}^{T}$, and B is a coefficient matrix, $B={\left[B_{1},B_{2},\ldots ,B_{m}\right]}^{T}$, Bi=[1β1−ki,1β2−ki,…,1βm−ki], i=1,2,…,m.

At this point, ${\left\|\theta \right\|}_{2}\gg 0$ is satisfied, with ${\left\|\cdot \right\|}_{2}$ representing the modulus of θ. However, in practice, most of θ can be approximated to 0. This approximates the column vector as a sparse vector, thus transforming the phasor estimation problem into a compressed-sensing model-solving problem.

Before calculating θ, perform the Fourier transform on Formula (12) to obtain the frequency domain expression:(12)$$x(g)=\sum_{u=-U}^{U}\left\{\sum_{k=0}^{K_{h}}\theta [\frac{1}{k!}{\left(\frac{j}{2\pi }\right)}^{k}D^{\left(k\right)}(\mu )]\right\}$$
where g is the frequency domain spectral line number, g∈(0,N−1), N is the number of sampling points, x(g) is the discrete Fourier value of $x_{h}(t)$, D(μ) is the Dirichlet function, $D\left(\mu \right)=\sin\left(\pi N\mu \right)/N\sin\left(\pi \mu \right)$, and $D^{\left(k\right)}(\mu )$ is the *k*th differentiation of D(μ). Coefficient μ can be expressed as $\mu =2\pi g/N-uf_{1}/f_{s}$.

By letting x(g)=x, the compressed sensing Taylor–Fourier (CSTFM) model can be obtained:(13)$$x=W^{H}D\theta$$
where $W^{H}$ is an N-order orthogonal matrix, Matrix D is an over-complete dictionary, $f_{u}=r_{u}\Delta_{f}$, Δf is the frequency resolution, and $r_{u}$ is the frequency index. The introduction of $f_{u}$ can improve frequency resolution. Since θ is approximately sparse, it can be estimated by the CS model and solved by the following formula:(14)$$\xi =\underset{p}{\arg\min}{\left\|\theta \right\|}_{0}$$
(15)$${\left\|x-W^{H}D\theta \right\|}_{2}\leq \eta$$
where ${\left\|\theta \right\|}_{0}$ is the number of non-zero elements in θ, ξ is the minimum value of ${\left\|\theta \right\|}_{0}$, and η is the threshold value for solving θ.

## 3. Primal–Dual Splitting Method

According to the above-mentioned conditions, sampling point X is needed to estimate θ. Based on the above model, the problem can be transformed into the following convex-optimization problem:(16)$$\underset{\theta }{\min}\;F(\theta )+Z(L\theta )$$
where L is a linear transformation matrix and F(θ) represents the measurement error. There are two main sources of measurement error: environment errors and hardware errors. For environmental errors, we deploy the anti-interference capability test in Section 4.3. For the sampling error caused by hardware, the literature offers few methods for eliminating such errors. Different hardware produces different errors, and most algorithms will be affected by such errors. This is an uncontrollable factor, and the error range is within ±5% (sampling equipment error). The influence of this error can be ignored, while the influence of noise error is much larger than the error caused by hardware, and it cannot be ignored.

For the error of the collected data in the transmission process, the current power quality data mainly comes from the merging unit of the digital substation, and the signal is transmitted in the form of a message, so transmission errors can also be ignored.

Therefore, this paper mainly focuses on the error caused by noise interference.

Introducing measurement-error entropy to represent measurement uncertainty:(17)H(X/θ)=H(X)−H(θ)

H(X) represents the information entropy of the results obtained from the sampling:(18)$$H(X)=-\sum_{i=0}^{N-1}P(X_{i})\log_{2}P(X_{i})$$

H(θ) represents information entropy of the known signals:(19)$$H(\theta )=-\int_{-\infty }^{+\infty }p(\theta ,t)\log_{2}p(\theta ,t)d\theta$$
where $P(\theta_{i}),P(x,t)$ denote the probability of occurrence of state $\theta_{i},x$.

Measurement error entropy is used to represent the uncertainty or discreteness of the measured parameters due to various factors (e.g., environment, equipment, etc.) in the actual sampling process.

Z(Lθ) is the regular item and F and Z are the lower semi-continuous convex function.

To ease the solution process, Formula (16) can be expressed in the norm-regularized form:(20)$$\underset{\theta }{\min}\frac{1}{2}{\left\|W^{H}D\theta -x\right\|}_{2}^{2}+\lambda {\left\|\theta \right\|}_{TV}$$
where $W^{H}D\in R^{n\times n}$ is the system matrix, $x\in R^{m}$ is the sampled signal, λ is the regularization coefficient, and λ>0. $R^{n},R^{m}$ are n-dimensional and m-dimensional real Euclidean spaces. By constructing the discrete difference matrix M and selecting an appropriate convex function g, the equivalent form of Formula (20) can be obtained:(21)$$\underset{\theta }{\min}\frac{1}{2}{\left\|W^{H}D\theta -x\right\|}_{2}^{2}+\lambda g(M\theta )$$

When θ satisfies the following fixed-point equation, the optimal solution $\theta^{*}$ to the convex optimization problem can be obtained:(22)$$\left(\begin{array}{l} \theta^{*}\\ y^{*} \end{array}\right)={\left(I+\gamma (S+\bar{M})\right)}^{-1}(1-\gamma \nabla \bar{F})\left(\begin{array}{l} \theta^{*}\\ y^{*} \end{array}\right)$$
where $y^{*}=\partial H(M\theta^{*}),S(\theta ,y)=(\partial \theta_{c}(\theta ),\partial H^{*}(y))$$\bar{M}=\left(\begin{array}{cc} 0 & M^{T}\\ -M & 0 \end{array}\right)$, $\nabla \bar{F}(\theta ,y)=(\nabla F(\theta ),0)$, I is an identity matrix of order n×n.

From the fixed-point equation obtained in (22), the following sequence of iterations can be obtained for any γ>0:(23)$$\left(\begin{array}{l} \theta^{k+1}\\ y^{k+1} \end{array}\right)={\left(1+\gamma (S+\bar{M})\right)}^{-1}(I-\gamma \nabla \bar{F})\left(\begin{array}{l} \theta^{*}\\ y^{*} \end{array}\right)$$

To create the iterative algorithm converge and have an explicit solution, γ can be chosen as a special positive definite matrix:(24)γ=1τI−MT−M1σI−1
where the iteration parameter τ,σ can be any value, as long as τ>0,σ>0.

Then, iteration Equation (23) can be equivalent to:(25)$$\begin{array}{l} \left(\begin{array}{l} \frac{1}{\tau }\theta^{k}-M^{T}y^{k}\\ -M\theta^{k}+\frac{1}{\sigma }y^{k} \end{array}\right)-\left(\begin{array}{c} \nabla F(\theta^{k})\\ 0 \end{array}\right)\in \left(\begin{array}{c} \frac{1}{\tau }\theta^{k+1}-M^{T}y^{k+1}\\ -M\theta^{k+1}+\frac{1}{\sigma }y^{k+1} \end{array}\right)\\ +\left(\begin{array}{c} \partial \theta_{C}(\theta^{k+1})+M^{T}y^{k+1}\\ -M\theta^{k+1}+\partial H^{*}(y^{k+1}) \end{array}\right) \end{array}$$

Formula (26) is acquired from (25): (26)$$\left(\begin{array}{c} 0\\ 0 \end{array}\right)\in \left(\begin{array}{l} \theta^{k+1}-(\theta^{k}-\tau M^{T}y^{k}-\tau \nabla F(\theta^{k}))+\tau \partial \theta_{c}(\theta^{k+1})\\ y^{k+1}-(y^{k}+\sigma M(2\theta^{k+1}-\theta^{k}))+\sigma \partial H^{*}(y^{k+1}) \end{array}\right)$$

The proximity operator and the projection operator are introduced to solve Equation (26):

Assuming that $R^{n}$ is an n-dimensional real Euclidean space, and C is a non-empty closed convex subset in $R^{n}$, the projection operator from any point θ in $R^{n}$ to C is:$P_{C}(\theta )=\arg\underset{y\in C}{\min}\left\|\theta -y\right\|$.

Similarly, it is assumed that f is a real-valued convex function defined on $R^{n}$, and for any λ>0,u∈R, the proximity operator is defined as: $prox_{\lambda f}(u)=\arg\underset{\theta }{\min}\left\{\frac{1}{2}{\left\|\theta -u\right\|}_{2}^{2}+\lambda f(\theta )\right\}$.

According to the definitions of the proximity operator and the projection operator, the specific iterative algorithm can be obtained through Equation (26):(27)$$\begin{array}{l} \theta^{k+1}=P_{C}(\theta^{k}-\tau M^{T}y^{k}-\tau \nabla F(\theta^{k}))\\ y^{k+1}=prox_{\sigma H^{*}}(y^{k}+\sigma M(2\theta^{k+1}-\theta^{k})) \end{array}$$
where the initial value of θ,y is arbitrary, and when the iteration stop condition ${\left\|W^{H}D\theta -x\right\|}_{2}\leq \eta$ is satisfied, $\theta^{k+1}$ is output.

Therefore, the parameter $\hat{\theta }$ can be obtained by an iterative method based on the primal–dual splitting method. Once $\hat{\theta }$ is determined, $X_{u},\varphi_{u},f_{u},R_{u}$ can be determined by the following formula. Once $X_{u},\varphi_{u},f_{u},R_{u}$ is determined, the reconstruction estimation of the dense signal can be completed:(28)Xu=2θ^ru(0)φu=arctanIm(θ^ru(0))Re(θ^ru(0))fu=ruΔf+Imθ^ru(1)θ^ru(0)*2π⋅θ^ru(0)2Ru=(θ^ru(2)θ^ru(0)−(θ^ru(1))2)2π(θ^ru(0))2
where ${\hat{\theta }}_{ru}$ is the $r_{u}th$ column of $\hat{\theta }$, ${\hat{\theta }}_{ru}^{\left(0\right)},{\hat{\theta }}_{ru}^{(0)*},{\hat{\theta }}_{ru}^{\left(1\right)}$ and ${\hat{\theta }}_{ru}^{\left(2\right)}$ are the zero-order derivative of ${\hat{\theta }}_{ru}$, the conjugate of the zero-order derivative, the first-order derivative, and the second-order derivative, respectively.

In summary, the specific steps of the dense signal-detection method based on the primal–dual splitting method are shown in Table 1:

## 4. Simulation

In this section, the basic performance of the primal–dual splitting method (PDSM) is tested via simulation experiments, and the frequency ramp and noise interference are selected as the test scenarios for the simulation experiments.

Total vector error (TVE), absolute frequency error (FE), and the absolute rate of change of frequency error (RFE) are introduced as evaluation indicators, comprehensively reflecting the error effects of amplitude, phase, and time, and describing the deviation between the theoretical phase and the estimated phase.

Experiments under different test conditions have been conducted to compare the accuracy of the proposed method with the DFT algorithm, Prony algorithm, TFT algorithm, and TWLS algorithm.

### 4.1. Basic Algorithm Performance

In this section, the PDSM is tested by simulation experiments. It is assumed that a single-channel signal consisting of multiple-frequency components can be represented as:(29)$$x(t)=\sum_{i=1}^{6}A_{i}\sin(2\pi f_{i}t+\varphi_{i}(t))$$
where $A_{i},f_{i},\varphi_{i}$ respectively represent the amplitude, frequency, and phase of the fundamental wave, as well as each harmonic and interharmonic component. The high percentage of nonlinear loads connected to the power grid and new energy generation connected to the grid generate such signals in practical applications. A typical example is the harmonics caused by electric arc furnaces. Ref. [37] indicates that the worst-case percentage of the second-harmonic component and the third-harmonic component are 17% and 29%, respectively. In addition, ref. [38] has pointed out that, when the furnace transformer is electrified, a large number of second-harmonic waves will be generated in the medium-voltage line. Based on the above factors, this paper selects the second and third harmonics and the adjacent interharmonic components as the main components for analysis. When i=1, it represents the fundamental wave component. While i=4 and i=6, it represents the second-harmonic component, and the third-harmonic component, and the others are the interharmonic components. The simulation signal parameters are shown in Table 2.

Total phasor error and iteration speed can intuitively reflect the performances of the algorithms. The iteration time and accuracy of PDSM are largely influenced by the iteration parameters τ,σ. By choosing the appropriate iterative parameters, the method has better and faster convergence. When τ<0 or σ<0, γ is not a positive definite matrix. Therefore, the calculation process does not converge, and the total vector error of the method approaches infinity, undermining the reliability of the result. When τ,σ is too large, the convergence speed of the method significantly slows down.

When the iteration parameters τ,σ vary, the analysis results of the total vector error and the running duration of the algorithm are as shown in Figure 1.

Figure 1 indicates that the convergence speed is fast when τ,σ are small, but the method produces a large total vector error in the meantime. Figure 2 shows more clearly how TVE changes with iterative parameters using a two-dimensional color map. As the iteration parameters τ,σ increase, the total vector error tends to first decrease and then increase. However, the running time of the method keeps increasing. When the chosen iteration parameters are too large, the iteration time of the method increases and so does the total vector error. When the iteration parameter τ∈(40,50), σ∈(8,9), the method has higher iteration accuracy and a faster iteration speed, achieving a better reconfiguration effect. Taking into account both the reconstruction performance and the operation time, this paper takes τ=44, σ=8.1. In this paper, the specified parameters are selected from the range class to construct the fixed-point equation, which is beneficial to the execution of the algorithm. For other signals, the convergence of the algorithm can be guaranteed, as long as τ,σ are selected within the “optimal” range, but the number of iterations required to obtain the convergence result is different. Total phasor error (TVE), frequency error (FE), and the rate of change of frequency error (RFE) are selected as evaluation indicators. The simulations compare the estimation accuracy of fundamental frequencies and each harmonic signal using PDSM, with the results obtained by the Prony method, TWLS method, TFT method, and DFT method. The comparison results are shown in Table 3.

As can be seen from Table 3, the maximum values of the TVE, FE, and RFE indexes of PDSM are 1.41%, 0.063 Hz, and 0.407 Hz/s, respectively, which meet the IEEE accuracy requirement for 1.78%, 0.092 Hz, and 0.48 Hz/s [39].

PDSM first transforms the time-domain model of the signal into the frequency domain by Fourier transformation to reduce the error of the method, then it uses the sparsity of the vector θ to construct a compressed perceptual model and transforms it into a convex-optimization problem, using the primal–dual splitting method to iterate over the fixed-point equations containing vector θ to further improve the accuracy of the method. When the iteration parameters τ,σ are chosen appropriately, the iteration converges well, which can tremendously reduce the iteration time while maintaining accuracy.

In this paper, the data in Table 3 were processed to improve the readability of the results. In the paper, green-labeled data are used to represent the minimum TVE, FE, and RFE for each component. Red-labeled data are used to represent the maximum TVE, FE, and RFE for each component. By comparing the data, it can be seen that, for each harmonic component, PDSM has a smaller TVE, FE, and RFE, while the TVE, FE, and RFE of the DFT algorithm are larger than other algorithms.

The DFT has low estimation accuracy due to spectral leakage and the fence effect, and the maximum TVE, FE, and RFE of the DFT algorithm are 2.654%, 0.121 Hz, and 0.912 Hz/s, respectively, which are much larger than the IEEE standards. The Prony algorithm transforms the time-domain analysis of the signal to the frequency domain and does not require the whole-period sampling of the signal, so the detection effect is better than the DFT algorithm. However, this method is prone to noise signals, and the detection effect is compromised when it comes to detecting a large number of harmonics and interharmonics. The maximum TVE of the Prony algorithm is 1.72%, and the maximum FE is 0.082 Hz, meeting the IEEE accuracy requirement, but the maximum RFE of the Prony algorithm is 0.521 Hz/s, which does not meet the requirement of IEEE. As a full extension of DFT, the TFT algorithm is based on the McLaughlin series expansion of each complex envelope. Since the Fourier subspace is contained in the subspace generated by the Taylor–Fourier basis, the estimation of the signal by the TFT algorithm has less distortion and less interharmonic interference. The maximum FE of the TFT algorithm is 0.0878 Hz, which meets the IEEE’s precision requirements, but the maximum TVE and maximum RFE of the TFT algorithm are 1.85% and 0.607 Hz/s, respectively, which are still too large compared to the IEEE standards. The TWLS algorithm uses the second-order Taylor series to fit the signal components, and the model has high accuracy with a maximum TVE and FE of 1.53% and 0.0684 Hz, respectively, meeting the IEEE requirement. However, the inherent expansion order of this method’s Taylor model causes errors in the reconstruction process, resulting in a maximum RFE of 0.512 Hz/s, which does not meet the IEEE accuracy requirements.

It can be concluded that PDSM fully meets the IEEE standards. In addition, comparison with other algorithms makes it clear that the estimation accuracy of PDSM is higher, and the maximum TVE, FE, and RFE of PDSM are smaller than those of the other, compared algorithms, which shows its superiority.

### 4.2. Frequency Ramp Test

The power imbalance between the load and the generator causes the frequency of the dense signal to decrease with increases in load. Additionally, the frequency increases when the input power is raised. To analyze the performance of PDSM with the frequency ramp, the signal tested is as follows:(30)$$x(t)=\sum_{i=1}^{6}A_{i}\sin(2\pi f_{i}t+\varphi_{i}(t))+0.1\sum_{h=2}^{H}\cos(2\pi hf_{0}t+\pi hRit^{2}+\varphi_{h}(t))$$
where $f_{0}$, having the value of 50 Hz, is the fundamental frequency. $R_{1}$, with the value 1 Hz/s, is the fundamental frequency ramp. The ramp and phase of each frequency component of the specific test signal are shown in Table 4.

The estimated results of TVE, FE, and RFE of PDSM and the compared algorithms are illustrated in Figure 2.

From Figure 3, it can be seen that, with the increase of the modulation frequency, the estimation accuracy of each algorithm will decrease. The maximum TVE, FE, and RFE of the proposed method in this paper are 1.57%, 0.172 Hz, and 1.308 Hz/s, respectively. The maximum TVE, FE, and RFE of the TWLS algorithm are 2.32%, 0.337 Hz, and 1.555 Hz/s, respectively. The maximum TVE, FE, and RFE of the TFT algorithm are 2.74%, 0.425 Hz, and 1.802 Hz/s, respectively. The maximum TVE, FE, and RFE of Prony’s algorithm are 2.52%, 0.376 Hz, and 1.631 Hz/s, respectively. The maximum TVE, FE, and RFE of the DFT algorithm are 3.19%, 0.56 Hz, and 3.024 Hz/s, respectively. The IEEE requirements for the maximum TVE, FE, and RFE under frequency ramp test conditions are 2.36%, 0.24 Hz, and 1.422 Hz/s. The primal–dual splitting algorithm is based on the fixed-point iterative equation when solving the optimal solution of the convex-optimization problem, and the appropriate iterative parameters are selected to make the matrix a special positive definite matrix, ensuring a good level of convergence of the iterative equation. When the frequency changes, the error can be continuously reduced through finite iterations. Therefore, PDSM demonstrates a good level of performance under the condition of the frequency ramp test and can accurately estimate the signal.

The TWLS algorithm uses the second-order Taylor series to fit the signal, so the effect of the linear frequency change on the algorithm is relatively small. However, the inherent expansion order of the Taylor model leads to errors when the model is reconstructed. When the signal frequency shifts, the TWLS algorithm cannot estimate the phase angle well. Although the TVE of the TWLS algorithm is relatively large, it still meets the IEEE standard. The TFT algorithm adopts the recursive calculation technology based on a multi-resonator after the second-order Taylor expansion. Due to its recursive form, the estimation accuracy of the algorithm is further improved. Compared with the DFT method, the Prony algorithm analyzes in the frequency domain, which reduces the error significantly, but the accuracy of these two methods still does not meet the IEEE requirements. When the frequency ramp changes, the DFT algorithm cannot track the change of the frequency in real-time. With the change of frequency, the frequency estimation error will gradually increase, and the DFT model cannot accurately track the change of the phasor in the observation window. Therefore, there will be a large error. The maximum TVE, FE, and RFE obtained from the measurement are much larger than the accuracy required by the IEEE, and the DFT algorithm has the lowest estimated accuracy among the five comparative calculations.

### 4.3. Anti-Interference Test

In this part of the test, high-frequency white noises with a signal-to-noise ratio of 55 dB and certain harmonic components are added to the signal. The specific dense signals are as follows:(31)$$x(t)=\sum_{i=1}^{6}A_{i}\sin(2\pi f_{i}t+\varphi_{i})+0.1\cos(2\pi hf_{t}t)+\mathrm{noise}$$

$f_{t}$ is the harmonic frequency, with values of 150 Hz, 200 Hz, 250 Hz, 300 Hz, 350 Hz, 400 Hz, and 450 Hz. The sampling frequency is 10 kHz, and the sampling window length is set to 5 frequency periods. The detailed values of other parameters in the test signal are shown in Table 5.

Data marked in green represent the minimum TVE, FE, and RFE for each component. Data marked in red represent the maximum TVE, FE, and RFE for each component. By comparing the data, it can be seen that, for each harmonic component, PDSM has a smaller TVE, FE, and RFE, while the TVE, FE, and RFE of the DFT algorithm are larger than those of other algorithms.

It can be seen from Table 5 that, under the test conditions of noise interference, the maximum TVE of the algorithm proposed in this paper is 2.21%, the maximum FE is 0.462 Hz, and the maximum RFE is 2.09 Hz/s. The maximum TVE, FE, and RFE of the TWLS algorithm are 2.83%, 0.612 Hz, and 2.54 Hz/s, respectively. The maximum TVE, FE, and RFE of the TFT algorithm are 5.78%, 1.321 Hz, and 5.38 Hz/s, respectively. The maximum TVE, FE, and RFE of the Prony algorithm are 3.69%, 1.074 Hz, and 3.932 Hz/s, respectively. The maximum TVE, FE, and RFE of the DFT algorithm are 6.84%, 1.532 Hz, and 6.93 Hz/s, respectively. In the case of noise interference, the IEEE’s maximum TVE, FE, and RFE values are 1.64%, 0.124 Hz, and 2.32 Hz/s, respectively. It can be seen from the data in Table 5 that only PDSM meets the IEEE standards. This is due to the fact that PDSM can strengthen the normalization constraints of the reconstruction problem by using the correlation between the coefficient matrices when building the compressed sensing model. The solution space is reduced, and noises are effectively suppressed, thus achieving the highest measurement accuracy.

Both the TWLS algorithm and the TFT algorithm are based on the second-order Taylor model to reconstruct the signal, which is more accurate than the DFT algorithm. However, the inherent expansion order of the Taylor model causes errors in the reconstruction process, and the noise interference in the TFT algorithm leads to severe spectrum aliasing between adjacent harmonics. Only the primary harmonics are kept in the waveform model, negatively affecting the estimation accuracy of the TFT algorithm. In the case of noise interference, the TWLS algorithm will increase the amplitude of the transition zone of the harmonic filter, resulting in spectrum aliasing between adjacent harmonics and a large estimation error. The Prony algorithm is highly sensitive to the noise in the signal and is prone to numerical instability. Therefore, when noise exists in the signal, it fails to make a good estimation of the dense signal. The DFT algorithm is most affected by noise interference since it has both serious spectrum leakage and the fence effect, and thus it is unable to accurately estimate the signal when noise interferes with it.

## 5. Conclusions

For the problem that dense-frequency signals are difficult to detect, this paper proposes a dense-signal-detection method based on the primal–dual splitting method. This paper first establishes the Taylor–Fourier model of the signal, uses the sparse characteristics of the coefficient matrix to obtain the CS model, and further expresses the CS model as a convex-optimization form. Then the primal–dual splitting method is used for the iteration to obtain the optimal solution of the convex-optimization model. Measurement error entropy is introduced during the process to describe the uncertainty of measurement, making the solution of the model more rigorous. The simulation and experimental results show that the proposed method can effectively improve the estimation accuracy of the phase. The method can meet the M-type PMU test requirements in frequency ramp and noise interference test scenarios. The method for performing the real-time detection of dynamic dense signals will be the next research direction.

## Figures and Tables

**Figure 1 entropy-24-00991-f001:**
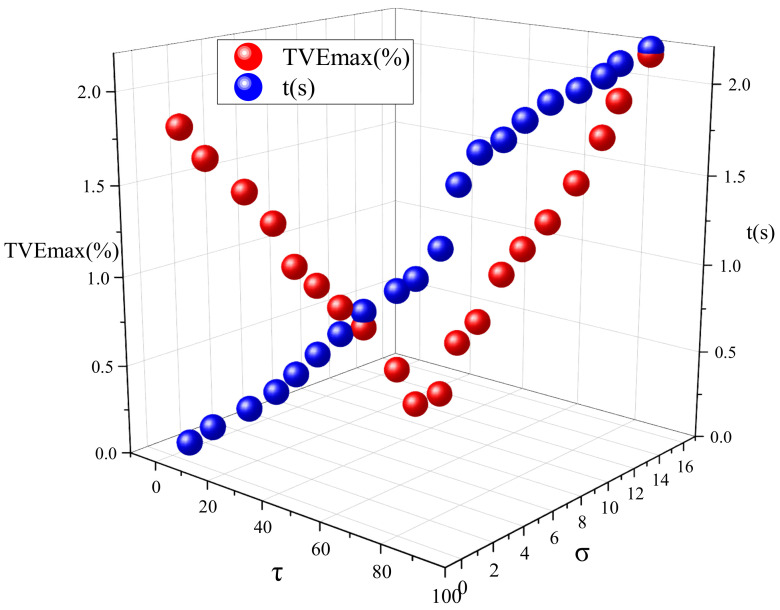
Total vector error and operation time with different iteration parameters.

**Figure 2 entropy-24-00991-f002:**
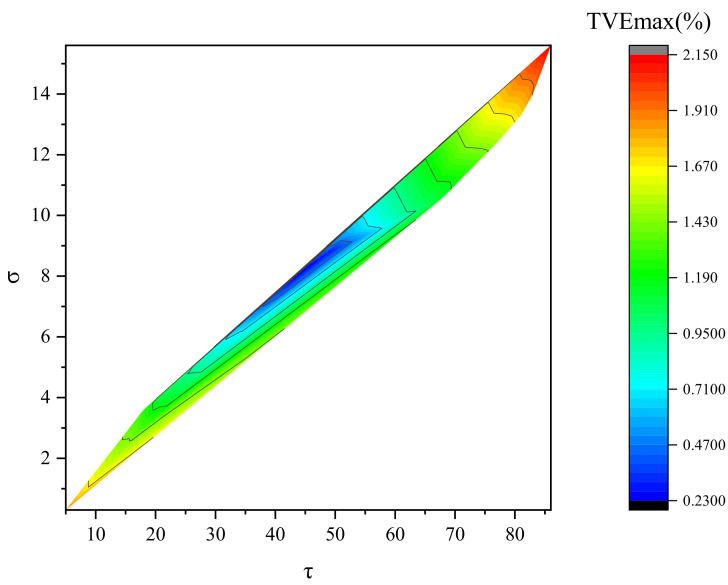
Total vector error with different iteration parameters.

**Figure 3 entropy-24-00991-f003:**
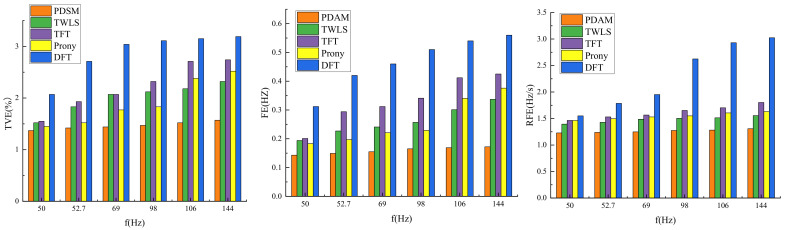
The maximum TVE, FE, and RFE of different methods under the frequency ramp condition.

**Table 1 entropy-24-00991-t001:** The specific steps of for implementing the method.

Initialization	
1	Input sample data X(k).
2	Complete Fourier transform on the model by Equation (13) to obtain the frequency domain expression, and CS reconstructs the model using Equation (14) to obtain initial estimates.
3	$\mathrm{Input}\;\mathrm{the}\;\mathrm{over}-\mathrm{complete}\;\mathrm{dictionary}\;D,\;\mathrm{Matrix}\;W^{H},\;\mathrm{and}\;\mathrm{set}\;\mathrm{the}\;\mathrm{threshold}\;\eta$ for iteration stop.
4	Transform the CS model into a convex optimization model and further express it as a norm-regularized form.
5	Construct the discrete difference matrix M, select the appropriate convex function g, and construct the fixed-point iterative equation.
6	$\mathrm{Select}\;\mathrm{Special}\;\mathrm{Positive}\;\mathrm{Definite}\;\mathrm{Matrix}\;\gamma ,\;\mathrm{iterative}\;\mathrm{parameter}\;\tau ,\sigma ,\;\mathrm{projection}\;\mathrm{operator}\;P_{C}(\theta ),\;\mathrm{and}\;\mathrm{proximity}\;\mathrm{operator}\;prox_{\lambda f}(u)$.
7	Obtain the optimal solution of θ by iteration.
8	$\mathrm{Obtain}\;X_{u},\varphi_{u},f_{u},R_{u}$ by Formula (25).

**Table 2 entropy-24-00991-t002:** Signal parameters.

I	1	2	3	4	5	6
$f_{i}\;\left(Hz\right)$	50	52.7	69	98	106	144
$A_{i}\;\left(\%\right)$	100	2.2	2.8	2.1	2.7	9.6
$\varphi_{i}\;\left(rad\right)$	0.38	0.26	0.33	0.36	0.48	0.24

**Table 3 entropy-24-00991-t003:** Comparison of the Estimation Accuracy of Algorithms.

i	TVE (%)					FE (Hz)					RFE (Hz/s)				
	PDSM	TWLS	TFT	Prony	DFT	PDSM	TWLS	TFT	Prony	DFT	PDSM	TWLS	TFT	Prony	DFT
1	1.23	1.31	1.42	1.36	1.682	0.058	0.065	0.0812	0.067	0.0852	0.34	0.389	0.447	0.412	0.671
2	1.29	1.35	1.46	1.38	1.751	0.059	0.067	0.0827	0.068	0.0874	0.36	0.407	0.483	0.445	0.731
3	1.34	1.37	1.54	1.41	1.863	0.06	0.0673	0.0841	0.069	0.0891	0.38	0.441	0.557	0.481	0.781
4	1.37	1.39	1.67	1.53	1.934	0.061	0.0676	0.0847	0.073	0.1072	0.392	0.467	0.573	0.493	0.827
5	1.38	1.45	1.76	1.61	2.137	0.062	0.0678	0.0859	0.075	0.1123	0.395	0.497	0.589	0.512	0.893
6	1.41	1.53	1.85	1.72	2.654	0.063	0.0684	0.0878	0.082	0.1211	0.407	0.512	0.607	0.521	0.912

**Table 4 entropy-24-00991-t004:** Ramp and phase of each frequency component.

I	$R_{i}$	$\varphi_{i}\;\left(rad\right)$
1	1	0.38
2	1.2	0.26
3	1.4	0.33
4	1.6	0.36
5	1.8	0.48
6	2.0	0.24

**Table 5 entropy-24-00991-t005:** Errors of each algorithm under noise interference.

i	TVE (%)					FE (Hz)					RFE (Hz/s)				
	PDSM	TWLS	TFT	Prony	DFT	PDSM	TWLS	TFT	Prony	DFT	PDSM	TWLS	TFT	Prony	DFT
1	1.87	2.14	3.17	2.74	4.12	0.391	0.421	0.674	0.581	0.874	1.74	1.96	2.437	2.315	3.74
2	1.93	2.25	3.46	2.86	4.34	0.407	0.487	0.831	0.743	0.943	1.78	1.98	2.943	2.873	4.21
3	1.97	2.41	3.82	3.04	4.56	0.413	0.512	0.942	0.792	1.108	1.94	2.07	3.67	2.941	4.83
4	2.05	2.56	4.13	3.26	4.78	0.421	0.536	1.03	0.835	1.137	2.02	2.27	4.12	3.321	5.42
5	2.14	2.67	5.31	3.48	5.97	0.453	0.579	1.132	0.943	1.254	2.06	2.43	4.57	3.574	6.31
6	2.21	2.83	5.78	3.69	6.84	0.462	0.612	1.321	1.074	1.532	2.09	2.54	5.38	3.932	6.93

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
