# Peer review of "Dense-Frequency Signal-Detection Based on the Primal–Dual Splitting Method"

_entropy, 2022, doi:10.3390/e24070991_

Round 1

Reviewer 1 Report

First of all revise the writing results because some mistakes call attention. For instance, at the beginning of the last parragraph in the introduction: "dence".  Sould be "dense". The matthematical formulation is very well deployed. Also, the general formulation of the research. Pleae, my question is that the paper lacks of a systematic error formulation related not only to the computation but also to the possible error sources in the data acquisition chain. Thanks

Reviewer 2 Report

The paper proposes a dense signal detection method based on the primal-dual splitting method. The general idea of the paper is ok. The proposed method is verified with simulations. Some questions and suggestions to improve the paper:

1) The authors state that a good choice of parameters \tau and \sigma is important for convergence. It seems that the values used were chosen based on figure 1. Are the chosen parameters also "optimal" for different signal parameters or do they have to be chosen each time? Furthermore to correctly visualize the values of TVE for different \tau and \sigma a 2d colour plot would be more informative.

2) An example signal is given for simulation tests of the proposed method. Does the proposed signal occur in a real-life application?

3) In Table 3: mark the best and worst estimation accuraccy for each case with color (green and red for example) to improve readability of the results

4) The  3d bar plots presented in Figure 2 are not very informative. A 2d plot (TVE(%) vs. frequency(Hz)) with all methods on one plot and each method marked with different color would be more clear.

5) Please proofread the paper there are some minor spelling mistakes. One example is Figure 2 where "frequence" should be replaced with "frequency". There are more such minor mistakes in the text.

Round 2

Reviewer 1 Report

paper has been improved according to my comments

Reviewer 2 Report

The authors have addressed all my remarks correctly. The paper can be accepted in its revised form.